# The Anti-Obesity Effect of Polysaccharide-Rich Red Algae (*Gelidium amansii*) Hot-Water Extracts in High-Fat Diet-Induced Obese Hamsters

**DOI:** 10.3390/md17090532

**Published:** 2019-09-13

**Authors:** Tsung-Han Yang, Chen-Yuan Chiu, Ting-Jang Lu, Shing-Hwa Liu, Meng-Tsan Chiang

**Affiliations:** 1Department of Food Science, National Taiwan Ocean University, Keelung 20224, Taiwan; tony81309@hotmail.com; 2Medical and Pharmaceutical Industry Technology and Development Center, New Taipei City 24886, Taiwan; 3Institute of Food Science and Technology, National Taiwan University, Taipei 10617, Taiwan; tjlu@ntu.edu.tw; 4Graduate Institute of Toxicology, College of Medicine, National Taiwan University, Taipei 10051, Taiwan; 5Department of Pediatrics, College of Medicine, National Taiwan University Hospital, Taipei 10051, Taiwan; 6Department of Medical Research, China Medical University Hospital, China Medical University, Taichung 40402, Taiwan

**Keywords:** *Gelidium amansii* hot-water extracts, guar gum, orlistat, obesity, adipocytokines, lipoprotein lipase

## Abstract

This study investigated the anti-obesity effect of a polysaccharide-rich red algae *Gelidium amansii* hot-water extract (GHE) in high-fat (HF) diet-induced obese hamsters. GHE contained 68.54% water-soluble indigestible carbohydrate polymers. Hamsters were fed with a HF diet for 5 weeks to induce obesity, and then randomly divided into: HF group, HF with 3% guar gum diet group, HF with 3% GHE diet group, and HF with orlistat (200 mg/kg diet) group for 9 weeks. The increased weights of body, liver, and adipose in the HF group were significantly reversed by GHE supplementation. Lower plasma leptin, tumor necrosis factor-α, and interleukin-6 levels were observed in the GHE+HF group compared to the HF group. GHE also increased the lipolysis rate and decreased the lipoprotein lipase activity in adipose tissues. GHE induced an increase in the phosphorylation of AMP-activated protein kinase (AMPK) and the protein expressions of peroxisome proliferator-activated receptor alpha (PPARα) and uncoupling protein (UCP)-2 in the livers. The decreased triglyceride and total cholesterol in the plasma and liver were also observed in obese hamsters fed a diet with GHE. These results suggest that GHE exerts a down-regulation effect on hepatic lipid metabolism through AMPK phosphorylation and up-regulation of PPARα and UCP-2 in HF-induced obese hamsters.

## 1. Introduction

Obesity is a global health care issue that is associated with metabolic diseases such as type 2 diabetes and hyperlipidemia [1,2]. Obesity, especially increased visceral adipose tissue, is a major risk factor for cardiovascular disease [3]. Hypertrophic adipocytes increase the generation of inflammatory cytokines such as tumor necrosis factor-α (TNFα) and interleukin-6 (IL-6), which may cause chronic inflammation and increase the risk of developing cardiovascular disease [4]. Functional foods have been demonstrated to possess anti-obesity potential [5,6]. Therefore, functional foods or food ingredients against obesity may come to be recognized as a promising therapeutic strategy for prediabetes and metabolic diseases.

*Gelidium amansii* (GA) is edible red algae, which is widely distributed in Japan, Korea, China, and northeast Taiwan. The agar product (1,3-linked b-D-galactopyranose and 1,4-linked 3,6-anhydro-a-L-galactopyranose units) of GA [7] can be prepared to form a gel [8]. It has been shown that GA possesses hypoglycemic effects in diabetic rats [9] and diabetic patients [10]. A recent study showed that the hot-water extracts of GA (GHE), an agar-filtered product, can reduce plasma and liver lipids by decreasing cholesterol absorption and increasing bile acid and fecal fat excretion in an animal model [11]. The water soluble fibers in GA and GHE may contribute to reducing lipid accumulation in the liver and adipose tissues. On the other hand, ethanol extracts of GA have been found to inhibit lipid accumulation in a 3T3-L1 cell model [12], and decrease the weights of body and epididymal fat and the levels of serum total cholesterol (TC) and triglyceride (TG) in an obese mouse model [13]. These findings suggest that GHE possesses beneficial effects for the improvement of lipid metabolism. However, the detailed effect and mechanism of GHE on obesity still remain to be clarified.

Hamsters make a suitable animal model for studying lipid metabolism, because their cholesterol and bile acid metabolism resemble to that of humans [14,15]. High fat (HF) diets have been shown to promote obesity and hyperlipidemia [16,17]. In addition, guar gum (GG), which is rich in water-soluble fibers processed from the endosperm of the cluster bean, is known for its hypolipidemic and anti-obesity effects [18,19]. In the present study, we used hamsters fed with a HF diet as an obese animal model to investigate the comparative effects of GHE and GG on lipid metabolism and obesity, and evaluated its possible mechanism of action. Orlistat, which is known as a potent inhibitor of gastric and pancreatic lipase and acts to reduce dietary fat absorption [20], was used as a positive control.

## 2. Results

### 2.1. Analysis of Carbohydrate Content and Monosaccharide Composition

The hot-water extract preparation was a polysaccharide-rich fraction of GA and contained 68.54 ± 1.43% carbohydrate polymers. Galactose (86.0%) was the major monosaccharide of the water-soluble indigestible polysaccharide from GHE, with low amounts of fucose (8.3%), mannose (1.5%), xylose (1.1%), and acidic sugar, glucuronic acid (2.0%). Rhamnose and glucose were also only detected in a much lower amounts (<1%) (Table 1).

### 2.2. Effects of GHE on Body and Tissue Weights in HF Diet-Fed Hamsters

Supplementation of GHE, GG, or orlistat significantly decreased body weight in HF diet-fed hamsters (Figure 1 and Table 2). As shown in Figure 1, hamsters fed a HF diet for 9 weeks had significantly increased liver and adipose weights, which could be significantly reversed by GHE, GG, or orlistat supplementation. GHE supplementation had lower adipose tissue weights than those fed a diet with GG supplementation in HF diet-fed hamsters. In addition, orlistat, but not both GHE and GG, significantly increased the food intake in hamsters fed with a HF diet (Table 2).

### 2.3. Effects of GHE on Plasma, Liver, and Fecal Lipids and Adipocytokines in HF Diet-Fed Hamsters

Hamsters fed with a HF diet had a significant increase in the plasma TC and TG levels, which could be significantly reversed by GHE, GG, or orlistat supplementation (Table 3). Moreover, animals fed the HF diet significantly enhanced TC and TG contents in the liver, which could also be significantly reversed by GHE, GG, or orlistat supplementation (Table 4). GHE supplementation had lower hepatic triglyceride contents than those fed a diet with GG supplementation in HF diet-fed hamsters (Table 4). Both GHE and orlistat supplementation significantly increased the fecal TC and TG contents; GG supplementation significantly increased the fecal TG, but not TC, contents (Table 4).

As shown in Table 3, hamsters fed with the HF diet had a significant increase in plasma leptin, IL-6, and TNF-α levels, which could be significantly reversed by GHE, GG, or orlistat supplementation.

### 2.4. Effects of GHE on TG Contents and Lipolysis Rate and LPL Activity in Adipose Tissues and Lipid Metabolism-Related Protein Expressions in the Livers

Hamsters fed the HF diet significantly increased TG contents in the adipose tissue, which could be significantly reversed by GHE, GG, or orlistat supplementation (Figure 2). It is interesting to note that the GHE group had lower TG contents in the adipose tissue than those of the GG group.

Hamsters fed the HF diet showed a decreased lipolysis rate and increased LPL activity in the adipose tissues (Figure 3). Supplementation of GHE, GG, or orlistat in the diet significantly increased the lipolysis rate and decreased the accumulation of TG and LPL activity in the adipose tissues. Higher lipolysis rate was observed in animals treated with GHE as compared to the GG group (Figure 3A).

The protein expressions of phosphorylated adenosine monophosphate (AMP)-activated protein kinase (AMPK), peroxisome proliferator activated receptor (PPAR)-α, and uncoupling protein (UCP)-2 were significantly decreased in the livers of obese hamsters, which could be significantly reversed by GHE, GG, or orlistat supplementation (Figure 4).

## 3. Discussion

In this study, we demonstrate for the first time that supplementation of GHE in HF diet-induced obese hamsters effectively induces (1) a reduction in body and adipose tissue weights, (2) a decrease in plasma TC/TG levels and hepatic TC/TG accumulation and adipose TG content, (3) an increase in fecal TC/TG content, (4) adipose tissue lipolysis and fatty acid β-oxidation, and (5) AMPK-related signals.

Our previous study showed that hamsters fed a HF diet with 1.5% GHE for 6 weeks effectively activate the phosphorylation of AMPK and reduce the protein expressions of SREBP-1 and SREBP-2 in the livers, thereby decreasing the hepatic lipogenesis and lipid accumulation as compared with HF diet-fed alone group [11]. In the present study, hamsters were fed with a HF diet for 5 weeks to induce obesity. GHE (3%) supplementation for 9 weeks significantly induced an anti-obesity effect and an increase in the AMPK phosphorylation and PPAR-α and UCP-2 protein expression in obese hamsters. AMPK activation is known as a major therapeutic target for obesity [21], fatty liver [22], and hyperlipidemia [23]. Yoon et al. showed that an increase in the expression of phosphorylated AMPK protein led to an increase in the expression of PPARα protein and an increase in the oxidation of fatty acids [24]. PPARα regulates target genes associated with fatty acid oxidation and lipid metabolism such as UCP-2 [25]. Therefore, one of the mechanisms by which obese hamsters with guar gum and GHE treatment can reduce the lipid contents in the liver may be due to promoting the oxidation of fatty acids in the liver by increasing the activities of phosphorylated AMPK, PPARα, and UCP-2 in the liver.

Consumption of a high-fat diet leads to obesity and hyperlipidemia in hamsters. Obese hamsters fed guar gum, GHE, or orlistat significantly reduced body weight, adipose tissue weight, and plasma levels of TC and TG. Orlistat has been reported to be an inhibitor of small intestinal lipase that reduces dietary fat absorption, and is used to treat obesity [20]. GG is rich in water-soluble fiber, and ingestion of GG can decrease plasma and liver lipids by increasing bile acid excretion to decrease fat absorption [26,27,28]. Frias and Sgarbieri showed that GG contained approximately 75% water-soluble fiber [29]. The GG and GHE used in the present study contained approximately 82.5% and 68.6% water-soluble fiber, respectively [11]. Thus, one of the reasons that GG and GHE can improve plasma and liver lipid metabolism may be related to the high levels of water-soluble fiber.

Adipose tissue is known to be an important endocrine organ that generates adipocytokines to affect energy metabolism and insulin sensitivity [30]. Hypertrophic adipocytes produce large amounts of inflammatory factors such as leptin, TNF-α, and IL-6 [31]. Feeding a red algae (GA) diet has been shown to significantly decrease plasma TNF-α, leptin, and TC/TG levels, but not affect the weights of body, liver, and adipose tissue, in a high fructose diet-impaired glucose tolerance rat model [32]. In the present study, both GHE (hot-water extract of red algae) and GG could significantly decrease plasma leptin, TNF-α, IL-6, and TC/TG levels and the weights of body, liver, and adipose tissue in obese hamsters. These results indicate that both GHE and GG may regulate plasma adipocytokines by lowering adipose tissue weight in hamsters. However, GHE is more effective than GG in reducing adipose tissue weight and the TG contents in the liver and adipose tissue. Moreover, lipoprotein lipase (LPL) activity in adipose tissue has been suggested to be closely related to obesity formation [33]. The present study also found that both GHE and GG could enhance the phosphorylation of AMPK in the liver, increase the lipolysis rate, and decrease the accumulation of TG and the LPL activity in the adipose tissues of obese hamsters. Watt et al. have demonstrated that increased AMPK phosphorylation can promote hormone-sensitive lipase (HSL) activation in the adipose tissue [34]. GG has been shown to enhance the activation of AMPK in both the liver and adipose tissue of HF diet-fed mice [35]. Kang et al. also found that the feeding of a red algae ethanol extract could increase the AMPK phosphorylation in the liver of obese mice and increase the amount of HSL protein in the adipose tissue [36]. Therefore, our findings suggest that GHE supplementation may decrease LPL activity and increase the rate of lipolysis, which lowers triglyceride content in the adipose tissue thus reducing the adipose tissue weight.

In this study, we found that galactose (86.0%) was the major monosaccharide of the water-soluble indigestible polysaccharide from GHE. We propose that galactans, indigestible polysaccharides, is a major and potent component of the seaweed extract. A similar polysaccharide rich extract from Ceylon moss (*Gelidium amansii*) has been reported by Wi et al. (2009) [37]. The report indicated that the seaweed contains 53.4% carbohydrates, consisting of galactose (23.4%) and glucose (22.3%). The difference of monosaccharide composition with low glucose content is a result of digestion treatment of α-amylase and amyloglucosidase used in our protocols. Our results provide more information by using a sophisticated hydrolysis protocol to release glycosidic linkages with less degradation. The structural details of the galactans need to be elucidated in future studies.

## 4. Materials and Methods

### 4.1. Reagents and Solvents

Barium acetate, chloroform, heparin, hydrochloric acid (HCl), isoproterenol, methanol, N-tris-(hydroxymethyl)methyl-2-aminoethanesulfonic acid, phenol, p-nitrophenyl butyrate, sodium acetate, sodium hydroxide (NaOH), sulfuric acid, and trifluoroacetic acid (TFA) were obtained from Sigma-Aldrich (St. Louis, MO, USA). A cholesterol enzymatic kit (Audit Diagnostics, Cork, Ireland), a TG enzymatic kit (Audit Diagnostics, Cork, Ireland), a glucose detection kit (Audit Diagnostics, Cork, Ireland), and plasma TNF-α, IL-6, and leptin ELISA kits (Assay Design, Ann Arbor, MI, USA) were used. A radio immunoprecipitation buffer (Cell Signaling Technology, Beverly, MA, USA) and an enhanced chemiluminescence kit (PerkinElmer, Waltham, MA, USA) were used.

### 4.2. Hot-Water Extract of GA (GHE)

The dry material of GA was obtained from the market at Keelung, Taiwan. GA was stored at 4 °C until used. For preparing GHE, GA (20 g) was added to the deionized water (400 mL), and then autoclaved at 121 °C for 20 min. After cooling, the samples were filtered and lyophilized. The harvest weight of GHE was 5.71 g, which the recovery rate was about 28.5%. The general composition of GHE, determined by the method of the Association of Official Agricultural Chemists (AOAC) [38], involved moisture 6.5%, ash 4.6%, crude fat 0.25%, and crude protein 6.7%. Moreover, GHE contained 68.6% water-soluble dietary fiber, according to the analysis by the Food Industry Research and Development Institute, Hsinchu, Taiwan [38].

### 4.3. Analysis of Carbohydrate Content and Monosaccharide Composition

GHE was re-dissolved in the deionized water and digested by α-amylase at pH 6.0, 30 min, 95 °C; protease at pH 7.5, 30 min, 60 °C; and amyloglucosidase at pH 4.5, 30 min, 60 °C. After the enzymes digestion, polysaccharides were precipitated and collected with 4 volumes of 95% ethanol.

The carbohydrate content was determined as previously described [39]. D-galactose was used as the standard and dissolved in the deionized water to construct the standard curve. The sample (200 μL) and 5% (W/V) phenol solution (200 μL) were mixed. After the solution was mixed, 1 mL 18 M sulfuric acid was added into the solution. After the solution was mixed and cooled, the absorbance at 490 nm using a spectrophotometer (Helios Omega, Thermo Fisher Scientific, San Jose, CA, USA) was measured. The absorbance value was subjected to the standard curve and the total carbohydrate concentration of the solution was calculated.

The monosaccharide composition analysis was performed as previously described [40]. The polysaccharides samples were methanolyzed with anhydrous 2 M HCl in absolute methanol, in a hydrolytic tube at 70 °C for 12 h. After methanolysis, the methanolysis reagent in the resulting solution was evaporated, and the residue was then subjected to acid hydrolysis with 2 M TFA at 100 °C for 1.5 h. In the resulting solution, TFA was removed by washing with methanol until TFA was removed. The sugars in the hydrolyzate were analyzed by high performance anion-exchange chromatography with pulsed amperometric detection (HPAEC-PAD). In neutral monosaccharide analysis, the analyte was separated by CarboPac PA10 guard column (4 mm × 50 mm) coupled with CarboPac PA10 IC Columns (4 mm × 250 mm). Columns were eluted at a flow rate of 0.5 mL/min with 25 mM NaOH, containing 1 mM barium acetate. In acid monosaccharide analysis, the analyte was separated by CarboPac PA10 guard column (4 mm × 50 mm) coupled with CarboPac PA10 IC Columns (4 mm × 250 mm). Columns were eluted at a flow rate of 0.8 mL/min with 75 mM NaOH, containing 1 mM barium acetate and 150 mM sodium acetate.

### 4.4. Animals and Treatments

The adult male Syrian hamsters (6-week-old) were obtained from The National Laboratory Animal Center (Taipei, Taiwan). Hamsters were individually housed in stainless steel cages in a room (23 ± 1 °C and 40–60% relative humidity) with a 12-h light–dark cycle. Hamsters were acclimated for 1 week and fed a standard laboratory diet (5001 rodent diet, Lab Diet, PMI Nutrition International Inc., Brentwood, MO, USA). Hamsters were then divided into normal and obese groups. Hamsters of the obese group were fed with a high-fat (HF) diet for 5 weeks to induce obesity. The obese hamsters were then randomly divided into four experimental groups: HF group, HF with 3% GG diet group (GG group), HF with 3% GHE diet group (GHE group), and HF with orlistat (200 mg/kg diet) group (OL group). The experimental diet compositions are listed in Table 5. Each experimental group was fed the experimental diets for 9 weeks. The normal control group was fed a standard laboratory diet (5001 rodent diet). Animals were given *ad libitum* access to food and water. Body weight was detected every week and fecal samples were collected during the final 3 days in week 9. The fecal samples were dried and weighed. This animal study was approved by the Animal House Management Committee of the National Taiwan Ocean University. All animal experiments were performed in accordance with the guidelines for the care and use of laboratory animals, as issued by the National Laboratory Animal Center. GG powder (GGP 200 LI-MV) was obtained from Lotus International (Rajasthan, India). Orlistat was as a positive control that was obtained from Weidar Chemical & Pharmaceutical CO., LTD. (Taichung, Taiwan).

### 4.5. Collection of Blood and Tissue Samples 

At the end of the experiment, overnight-fasted hamsters were sacrificed under CO_2_ anesthesia. Blood was collected and plasma was prepared by centrifugation at 1750× *g* for 20 min (4 °C). Liver and perirenal and para-epididymal adipose tissues were isolated and weighed. All tissue samples were immediately frozen and stored at ‒80 °C until further analysis.

### 4.6. Determination of Plasma Lipids 

Both TC and TG levels in the plasma were determined by using a cholesterol enzymatic kit (Audit Diagnostics) and TG enzymatic kit (Audit Diagnostics), respectively.

### 4.7. Determination of Plasma Glucose, Tumor Necrosis Factor α (TNF-α), Interleukin 6 (IL-6), and Leptin

Plasma glucose was measured using a glucose detection kit (Audit Diagnostics, Cork, Ireland). The plasma levels of TNF-α, IL-6, and leptin were determined by the ELISA kits (Assay Design, Ann Arbor, MI, USA).

### 4.8. Determination of Liver Lipids, Fecal Lipids, and Bile Acid Levels

Both liver and fecal lipids were extracted with a chloroform/methanol solution (v/v, 2:1) as previously described by Folch et al. [41]. The levels of both TG and TC were determined by the method of Carlson and Goldfarb [42]. Fecal bile acids were extracted and detected according to the method of Cheng and Lai [43].

### 4.9. Lipolysis Rate Measurement

Lipolysis rate measurement was determined as previously described [44]. The 0.2 g of adipose samples were minced and placed into 2 mL of N-tris-(hydroxymethyl)methyl-2-aminoethanesulfonic acid (25 mM) buffer containing isoproterenol (1 μM), and then incubated at 37 °C. After 1, 2, and 3 h of incubation, 0.2 mL of medium was used to analyze the concentrations of glycerol using a commercial reagent (RANDOX GY105, Amtrim, UK), and then the absorbance at 520 nm was recorded by a Hitachi U2800A spectrophotometer. The lipolysis rate was indicated by micromoles of glycerol released per gram of tissue per hour.

### 4.10. Lipoprotein Lipase (LPL) Activity

LPL activity in adipose samples was determined as previously described [45]. The 0.1 g of adipose samples were minced and placed into Krebs-Ringer bicarbonate buffer (pH 7.4) containing heparin (10 units/mL) for 60 min at 37 °C. The heparin solution was mixed with an equal volume of p-nitrophenyl butyrate (pNPB; 2 mM). Absorbance at 400 nm was recorded by a Hitachi U2800A spectrophotometer. LPL activity was indicated as the amount of p-nitrophenol formed over the 10 min incubation.

### 4.11. Western Blot Analysis

The proteins in the livers were extracted using a radio immunoprecipitation buffer (Cell Signaling Technology). The 40 μg of tissue proteins were separated on 10% sodium dodecyl sulfate polyacrylamide gels and transferred to polyvinylidene fluoride membranes (Bio-Rad, CA, USA). After blocking, the membranes were probed with primary antibodies for β-actin (Biovision, Mountain View, CA, USA), AMPK, phospho-AMPK (Thr172) (Cell Signaling Technology), PPAR-α (Santa Cruz Biotechnology, Santa Cruz, CA, USA), and UCP-2 (Origene, Rockville, MD, USA) overnight at 4 °C, followed by incubation with horseradish peroxidase linked secondary antibodies for 1 h at room temperature, and then detected by enhanced chemiluminescence (PerkinElmer). Signal intensities were quantitated using the Quantity One Software (BioRad, Hercules, CA, USA).

### 4.12. Statistical Analysis

Data are presented as mean ± standard deviation. The significant difference from the respective controls for each experimental condition was assessed by one-way analysis of variance and two-tailed Student *t*-test. The statistical analysis was performed by statistical software SPSS for Windows version 12.0 (SPSS, Chicago, IL, USA).

## 5. Conclusions

Based on these findings, supplementation of GHE to HF diet-induced obese hamsters can activate AMPK phosphorylation and up-regulate PPAR-α and UCP-2 protein expressions, which increase the fatty acid β-oxidation in the liver. GHE has beneficial effects on reducing the weight of body and adipose tissue, plasma TC and TG levels, TG content in adipose tissue, and hepatic lipid accumulation.

## Figures and Tables

**Figure 1 marinedrugs-17-00532-f001:**
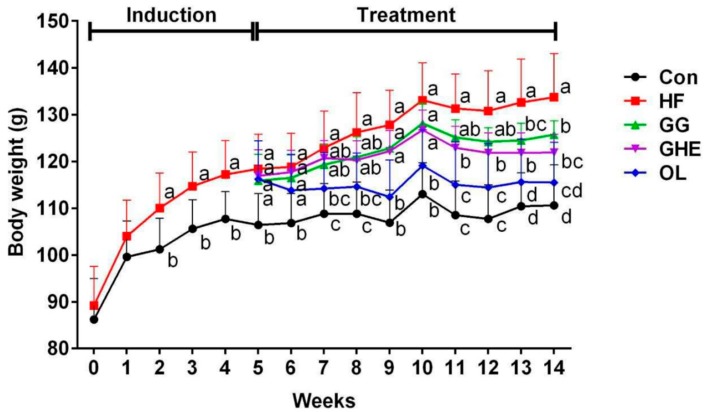
Effects of GHE on body weight in hamsters fed different diets for 14 weeks during induction period (weeks 0–5) and experimental period (weeks 6–14). Data are presented as mean ± SD (*n* = 8). Con: Normal control group; HF: Obese hamsters fed a high-fat diet; GG: Obese hamsters fed 3% guar gum in the high-fat diet; GHE: Obese hamsters fed 3% *Gelidium amansii* hot-water extract in the high-fat diet; OL: Obese hamsters fed a high-fat diet with orlistat (200 mg/kg diet) in the diet. Values denoted by different letters differ significantly (*p* < 0.05) among one another.

**Figure 2 marinedrugs-17-00532-f002:**
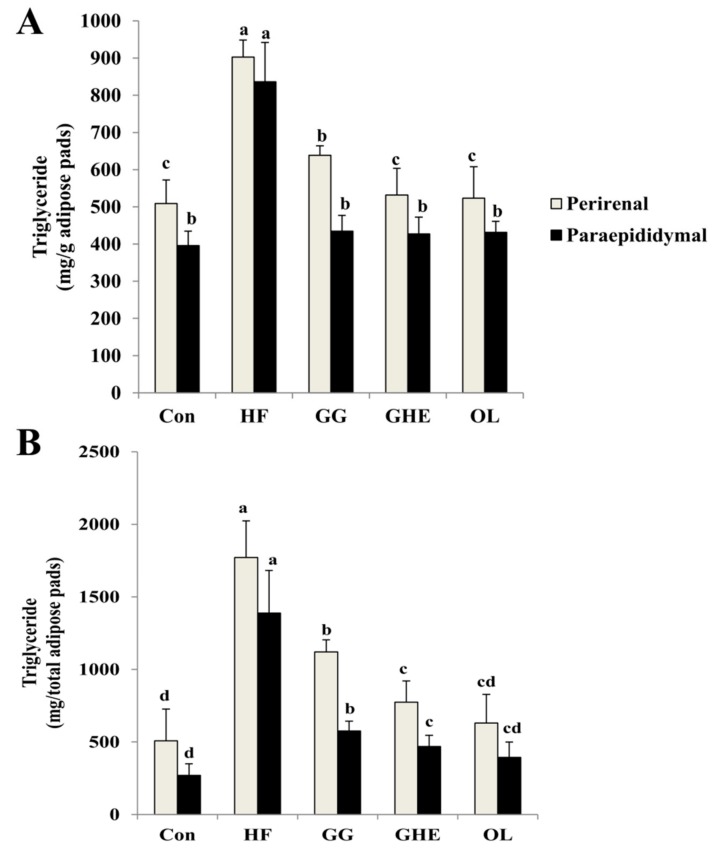
Effects of GHE on triglyceride content per gram (**A**) and total (**B**) paraepididymal and perirenal fat pads in hamsters fed different experimental diets for 9 weeks. Data are presented as mean ± SD (*n* = 8). Con: Normal control group; HF: Obese hamsters fed a high-fat diet; GG: Obese hamsters fed 3% guar gum in the high-fat diet; GHE: Obese hamsters fed 3% *Gelidium amansii* hot-water extract in the high-fat diet; OL: Obese hamsters fed a high-fat diet with orlistat (200 mg/kg diet) in the diet. Values denoted by different letters differ significantly (*p* < 0.05) among one another.

**Figure 3 marinedrugs-17-00532-f003:**
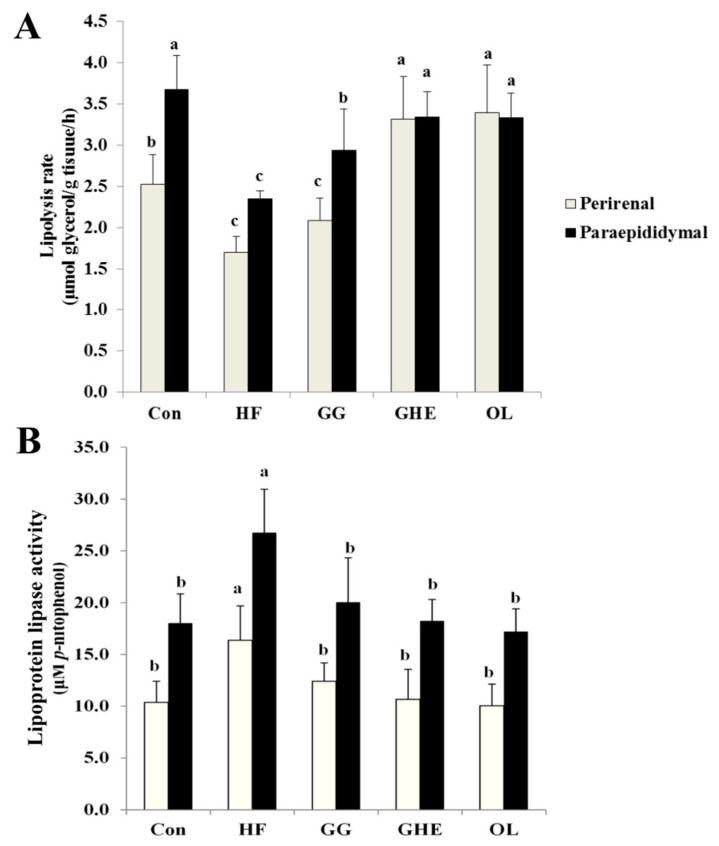
Effects of GHE on lipolysis rate (**A**) and lipoprotein lipase activity (**B**) of paraepididymal and perirenal fat pads in hamsters fed different experimental diets for 9 weeks. Data are presented as mean ± SD (*n* = 8). Con: Normal control group; HF: Obese hamsters fed a high-fat diet; GG: Obese hamsters fed 3% guar gum in the high-fat diet; GHE: Obese hamsters fed 3% *Gelidium amansii* hot-water extract in the high-fat diet; OL: Obese hamsters fed a high-fat diet with orlistat (200 mg/kg diet) in the diet. Values denoted by different letters differ significantly (*p* < 0.05) among one another.

**Figure 4 marinedrugs-17-00532-f004:**
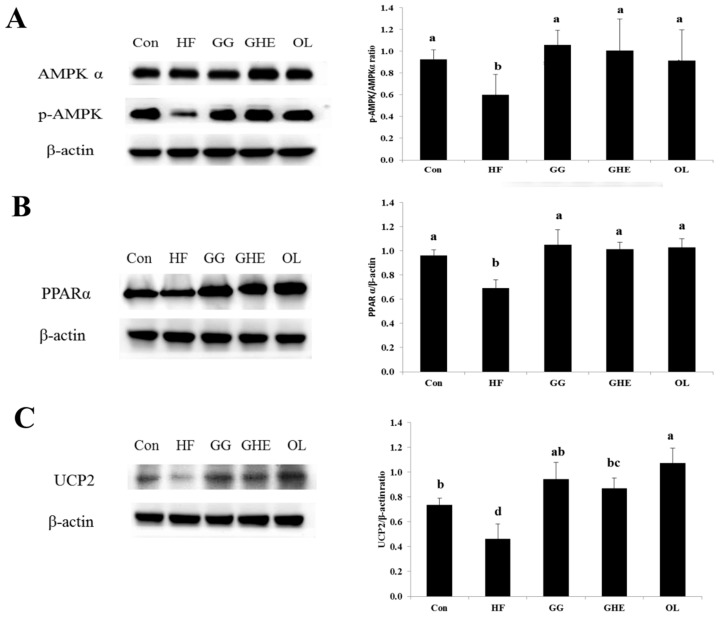
Effects of GHE on the protein expressions of phosphorylated AMPK (**A**), PPARα (**B**), and UCP-2 (**C**) in the livers of hamsters fed different experimental diets for 9 weeks. Data are presented as mean ± SD for 4–5 Syrian hamsters. Con: Normal control group; HF: Obese hamsters fed a high-fat diet; GG: Obese hamsters fed 3% guar gum in the high-fat diet; GHE: Obese hamsters fed 3% *Gelidium amansii* hot-water extract in the high-fat diet; OL: Obese hamsters fed a high-fat diet with orlistat (200 mg/kg diet) in the diet. Values denoted by different letters differ significantly (*p* < 0.05) among one another.

**Table 1 marinedrugs-17-00532-t001:** Monosaccharide composition of the water-soluble indigestible polysaccharide from the *Gelidium amansii* hot-water extract (GHE).

Molar Ratio of Monosaccharide Composition (100%)								
Fuc	Rha	Ara	Gal	Glc	Man	Xyl	GalA	GlcA
8.3 ± 0.1	0.5 ± <0.1	nd	86.0 ± 0.5	0.6 ± 0.1	1.5 ± <0.1	1.1 ± <0.1	nd	2.0 ± 0.1

Data are presented as mean ± SD (*n* = 3). Values were determined by high-performance anion-exchange chromatography with pulsed amperometric detection (HPAEC-PAD). Fuc: Fucose; Rha: Rhamnose; Ara: Arabinose; Gal: Galactose; Glc: Glucose; Man: Mannose; Xyl: Xylose; Gal A: Galacturonic acid; Glc A: Glucuronic acid; nd: Non-detected.

**Table 2 marinedrugs-17-00532-t002:** Effects of body weights, food intake and tissue weights in hamsters fed different diets for 9 weeks.

Parameters	Con	HF	GG	GHE	OL
Initial body weight (g)	106.4 ± 6.7 ^b^	118.4 ± 7.4 ^a^	115.9 ± 5.6 ^a^	116.9 ± 5.6 ^a^	116.2 ± 8.2 ^a^
Final body weight (g)	110.6 ± 8.7 ^d^	133.7 ± 9.4 ^a^	125.7 ± 3.0 ^b^	121.9 ± 3.9 ^bc^	115.5 ± 8.6 ^cd^
Body weight gain (g)	4.2 ± 7.9 ^bc^	15.3 ± 4.5 ^a^	9.8 ± 3.8 ^b^	5.0 ± 4.5 ^bc^	-0.7 ± 5.2 ^c^
Food intake (g/days)	8.3 ± 0.3 ^b^	8.1 ± 0.4 ^bc^	7.8 ± 0.4 ^c^	7.8 ± 0.4 ^c^	9.3 ± 0.3 ^a^
Feed efficiency ratio (%)	0.7 ± 1.4 ^cd^	2.7 ± 0.8 ^a^	1.8 ± 0.7 ^ab^	0.9 ± 0.8 ^bc^	−0.1 ± 0.8 ^d^
Fasting body weight (g)	108.3 ± 8.3 ^d^	130.6 ± 9.5 ^a^	123.5 ± 4.3 ^ab^	119.3 ± 3.8 ^bc^	113.2 ± 9.2 ^cd^
Liver weight (g)	3.2 ± 0.3 ^d^	6.1 ± 0.3 ^a^	5.1 ± 0.3 ^b^	5.0 ± 0.1 ^b^	3.6 ± 0.2 ^c^
Relative liver weight (g/100 g BW)	3.0 ± 0.2 ^d^	4.7 ± 0.2 ^a^	4.1 ± 0.1 ^b^	4.2 ± 0.1 ^b^	3.2 ± 0.2 ^c^
Perirenal fat (g)	1.0 ± 0.3 ^c^	2.0 ± 0.2 ^a^	1.8 ± 0.1 ^a^	1.5 ± 0.2 ^b^	1.2 ± 0.2 ^c^
Paraepididymal fat (g)	0.7 ± 0.2 ^d^	1.7 ± 0.2 ^a^	1.3 ± 0.1 ^b^	1.1 ± 0.2 ^c^	0.9 ± 0.2 ^c^
White adipose tissue weight (g)	1.7 ± 0.6 ^e^	3.6 ± 0.4 ^a^	3.1 ± 0.2 ^b^	2.6 ± 0.3 ^c^	2.1 ± 0.5 ^d^
Relative white adipose tissue weight (g/100 g BW)	1.5 ± 0.4 ^d^	2.8 ± 0.2 ^a^	2.5 ± 0.2 ^a^	2.1 ± 0.2 ^b^	1.8 ± 0.3 ^c^

Data are presented as mean ± SD (*n* = 8). Con: Normal control group; HF: Obese hamsters fed a high-fat diet; GG: Obese hamsters fed 3% guar gum in the high-fat diet; GHE: Obese hamsters fed 3% *Gelidium amansii* hot-water extract in the high-fat diet; OL: Obese hamsters fed a high-fat diet with orlistat (200 mg/kg diet) in the diet. Values denoted by different letters differ significantly (*p* < 0.05) among one another. Food efficiency ratio (FER) = [weight gain (g)]/[total food intake (g)] × 100. Values denoted by different letters differ significantly (*p* < 0.05) among one another.

**Table 3 marinedrugs-17-00532-t003:** Effects of different diets on plasma biochemistry in hamsters for 9 weeks.

Parameters	Con	HF	GG	GHE	OL
Glucose (mg/dL)	148.8 ± 13.6 ^a^	153.3 ± 12.5 ^a^	144.2 ± 21.1 ^a^	144.9 ± 23.1 ^a^	161.0 ± 15.7 ^a^
Triglyceride (mg/dL)	58.4 ± 13.9 ^c^	173.7 ± 34.7 ^a^	139.3 ± 27.6 ^b^	128.1 ± 21.2 ^b^	119.8 ± 28.1 ^b^
Total cholesterol (mg/dL)	80.6 ± 13.4 ^d^	390.1 ± 76.9 ^a^	252.4 ± 31.7 ^b^	285.9 ± 42.0 ^b^	136.4 ± 22.5 ^c^
Leptin (pg/mL)	245.4 ± 81.3 ^d^	818.6 ± 241.9 ^a^	607.9 ± 232.0 ^b^	484.0 ± 155.5 ^bc^	352.5 ± 134.2 ^cd^
TNFα (pg/mL)	37.7 ± 1.4 ^d^	53.8 ± 1.2 ^a^	40.5 ± 1.8 ^b^	39.2 ± 2.1 ^bc^	37.3 ± 0.7 ^cd^
IL-6 (pg/mL)	10.5 ± 1.5 ^b^	21.5 ± 8.2 ^a^	11.8 ± 3.2 ^b^	10.2 ± 3.4 ^b^	8.8 ± 2.1 ^b^

Data are presented as mean ± SD (*n* = 8). Con: Normal control group; HF: Obese hamsters fed a high-fat diet; GG: Obese hamsters fed 3% guar gum in the high-fat diet; GHE: Obese hamsters fed 3% *Gelidium amansii* hot-water extract in the high-fat diet; OL: Obese hamsters fed a high-fat diet with orlistat (200 mg/kg diet) in the diet. Values denoted by different letters differ significantly (*p* < 0.05) among one another.

**Table 4 marinedrugs-17-00532-t004:** Effects of different diets on liver and fecal lipid concentrations in Syrian hamsters for 9 weeks.

Lipids	Con	HF	GG	GHE	OL
**Liver**					
Total cholesterol (mg/g Liver)	5.9 ± 1.9 ^d^	71.2 ± 6.4 ^a^	42.8 ± 5.8 ^b^	40.4 ± 6.5 ^b^	17.0 ± 6.7 ^c^
Total cholesterol (mg/Liver)	19.1 ± 7.3 ^d^	434.1 ± 44.1 ^a^	218.3 ± 31.9 ^b^	201.2 ± 31.3 ^b^	61.0 ± 21.6 ^c^
Triglyceride (mg/g Liver)	2.9 ± 0.4 ^d^	9.5 ± 1.1 ^a^	7.8 ± 1.7 ^b^	6.1 ± 1.6 ^c^	7.3 ± 1.5 ^bc^
Triglyceride (mg/Liver)	9.2 ± 1.7 ^d^	57.6 ± 5.6 ^a^	39.4 ± 7.9 ^b^	30.6 ± 7.9 ^c^	26.6 ± 5.9 ^c^
**Feces**					
Total cholesterol (mg/g feces)	3.7 ± 0.5 ^d^	4.7 ± 0.9 ^cd^	5.0 ± 1.3 ^c^	10.3 ± 1.1 ^a^	7.1 ± 1.5 ^b^
Total cholesterol (mg/day)	4.3 ± 0.6 ^b^	5.5 ± 1.3 ^b^	5.3 ± 1.7 ^b^	11.8 ± 1.6 ^a^	12.5 ± 3.5 ^a^
Triglyceride (mg/g feces)	3.2 ± 0.2 ^c^	3.5 ± 0.3 ^c^	4.7 ± 0.5 ^b^	4.7 ± 0.3 ^b^	106.4 ± 13.8 ^a^
Triglyceride (mg/day)	3.8 ± 0.4 ^c^	4.0 ± 0.5 ^c^	5.0 ± 0.7 ^b^	5.5 ± 0.7 ^b^	185.5 ± 29.4 ^a^

Data are presented as mean ± SD (*n* = 8). Con: Normal control group; HF: Obese hamsters fed a high-fat diet; GG: Obese hamsters fed 3% guar gum in the high-fat diet; GHE: Obese hamsters fed 3% *Gelidium amansii* hot-water extract in the high-fat diet; OL: Obese hamster fed a high-fat diet with orlistat (200 mg/kg diet) in the diet. Values denoted by different letters differ significantly (*p* < 0.05) among one another.

**Table 5 marinedrugs-17-00532-t005:** Composition of experimental diets (%).

Composition	Con	HF	GG	GHE	OL
Chow diet	100	86.9	86.9	86.9	86.9
Coconut oil	-	6	6	6	6
Soybean oil	-	4	4	4	4
Cholesterol	-	0.1	0.1	0.1	0.1
Cellulose	-	3	-	-	3
*Gelidium amansii* hot-water extract	-	-	-	3	-
Guar gum	-	-	3		-
Total	100	100	100	100	100
Orlistat	-	-	-	-	200 mg/kg diet
Kcal (100 g)	345.6	396.3	395.9	396.9	396.3

Con: Hamsters fed a chow diet group; HF: Obese hamsters fed a high-fat diet group; GG: Obese hamsters fed 3% guar gum in the high-fat diet group; GHE: Obese hamsters fed 3% *Gelidium amansii* hot-water extract in the high-fat diet; OL: Obese hamsters fed a high-fat diet with orlistat (200 mg/kg diet).

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
