# Peer review of "The Anti-Obesity Effect of Polysaccharide-Rich Red Algae (Gelidium amansii) Hot-Water Extracts in High-Fat Diet-Induced Obese Hamsters"

_marinedrugs, 2019, doi:10.3390/md17090532_

Round 1
Reviewer 1 Report
1- line 90, it has mentioned that experimental period (weeks 5-14), but this will be 10 weeks in total not 9 weeks as mentioned before in line 82, please double check this. 2- In line 82, it mentioned that" As shown in Table 1, hamsters fed a HF diet for 9 weeks", however, table 1 title is the Monosaccharide composition of the water soluble indigestible polysaccharide from GHE, if the authors mean figure 1, please correct this.
Author Response
Reviewer 1:
1- line 90, it has mentioned that experimental period (weeks 5-14), but this will be 10 weeks in total not 9 weeks as mentioned before in line 82, please double check this.
Response: We appreciate the reviewer's comment. We have corrected the experimental period that is weeks 6-14 (the treatment period is 9 weeks).
2- In line 82, it mentioned that" As shown in Table 1, hamsters fed a HF diet for 9 weeks", however, table 1 title is the Monosaccharide composition of the water soluble indigestible polysaccharide from GHE, if the authors mean figure 1, please correct this.
Response: We appreciate the reviewer's comment. We have corrected this typing error that the figure 1 is correct.
Reviewer 2 Report
This paper describes the anti-obesity activity of hot-water extracts of Gelidium amansii in high-fat diet-induced obese hamsters in a satisfactory way. The work is interesting and informative. However, I have some minor observations that are listed in the following lines.
There are several grammatical and stylistic mistakes. This work would benefit from close editing. Figure 1: the error bars are quite high. Table 3: the SD for leptin results is really high in all groups. Line 226: The materials section should list the reagents and solvents used and their source. Line 236: Provide a reference for the method used for the analysis of carbohydrate content and monosaccharide composition. Lines 243, 306 and 312: specify the model of the spectrophotometer used.
Author Response
This paper describes the anti-obesity activity of hot-water extracts of Gelidium amansii in high-fat diet-induced obese hamsters in a satisfactory way. The work is interesting and informative. However, I have some minor observations that are listed in the following lines.
(1) There are several grammatical and stylistic mistakes. This work would benefit from close editing. (2) Figure 1: the error bars are quite high. Table 3: the SD for leptin results is really high in all groups. (3) Line 226: The materials section should list the reagents and solvents used and their source. Line 236: Provide a reference for the method used for the analysis of carbohydrate content and monosaccharide composition. Lines 243, 306 and 312: specify the model of the spectrophotometer used.
Response: We appreciate the reviewer's comment. (1) We have carefully checked and revised the manuscript according to the suggestion of reviewer. (2) We have carefully checked and statistically re-analyzed the data in Figure 1 and Table 3. The data are correct. (3) We have added a sub-section for reagents and solvents in Materials and Methods section of this revised manuscript according to the suggestion of reviewer. We also added the references for the methods of carbohydrate content and monosaccharide composition according to the suggestion of reviewer. We also added the descriptions for the model of the spectrophotometer used in the Materials and Methods section according to the suggestion of reviewer.